# Thermal Error Analysis of Five-Axis Machine Tools Based on Five-Point Test Method

Yu Li [1], Hongchuan Tian [1], Difei Liu [1] and Quanbo Lu [2,*]

[1] China Academy of Information and Communications Technology, Beijing 100045, China; liyu@caict.ac.cn (Y.L.); tianhongchuan@caict.ac.cn (H.T.); liudifei@caict.ac.cn (D.L.)
[2] School of Information Engineering, China University of Geosciences, Beijing 100083, China
* Correspondence: luquanbo111@163.com

**Abstract:** The accuracy of five-axis machine tools is a key performance indicator. Among the various error sources of high precision five-axis machine tools, thermal and geometric errors occupy the majority. Thermal errors have become the largest error source of high precision five-axis machine tools, accounting for about 45% of the total errors. Accurate measurement of thermal errors plays a vital role in improving the accuracy of five-axis machine tools. Taking the Shenyang HTM50100 turning and milling machine tool as an example, this paper proposes a method to measure the thermal error of the machine tool spindle using the five-point test method. In the process of thermal error modeling, we select the temperature key point and analyze the collected data. Finally, we evaluate thermal error model. The method is verified by an experiment. The experiment results show that the method is highly accurate, fast, and easy to use. It provides a theoretical basis and practical method for the measurement of thermal errors on five-axis machine tools. By evaluating the method based on multiple linear regression, the predictive ability of the model is about 77%. Compared with LSTM, the prediction accuracy is improved by 5.08%.

**Keywords:** five-axis machine tools; five-point test; thermal error; multiple linear regression

## 1. Introduction

Precision and ultra-precision machining technology has become the most important part of modern machinery manufacturing. It becomes a key technology to improve international competitiveness [1–3]. Among the various error sources of high precision five-axis machine tools, thermal and geometric errors occupy the majority. Thermal errors have become the largest error source of high precision five-axis machine tools, accounting for about 45% of the total errors [4–6]. The accuracy of five-axis machine tools is ultimately determined by the relative displacement between the tool and the workpiece on the five-axis machine tools. The integrated error between the tool and the workpiece (i.e., position and direction error) will affect the relative displacement between the tool and the workpiece. To improve the machining accuracy of five-axis machine tools, it must compensate for the thermal error of five-axis machine tools [7–11]. Gantry machines are well studied. E Gomez–Acedo proposed an approach for thermal characterization. It is the way to deal with non-linear problems [12]. The idea is that a parametric state-space representation was selected as model architecture, providing multiple inputs and outputs capability and a compact formulation that takes into account previous thermal states of the machine [13]. The group by Prof K Weneger actively worked on thermal models and compensation for milling centers and turn-milling machine improvement [14,15]. O Horejš proposed a continuation of scientific work on advanced modelling of thermally induced displacements based on thermal transfer functions [16]. S Ibaraki proposed a machining test, which is to evaluate the thermal influence on position and orientation errors of rotary axis average lines in a five-axis machine tool [17].

In the process of five-axis machine tools processing, the main heat source of five-axis machine tools includes the friction heat between the spindle and sleeve, environmental temperature, motor heating, friction heat between guide rail and bed. These heat sources will lead to the change of the relative position between the tool and the workpiece, resulting in complex thermal deformation errors. In order to realize the precision and high precision machining of five-axis machine tools, it is necessary to establish the mathematical model between the thermal error of five-axis machine tools and the sensitive heat source of machine tools [18–23]. It effectively predicts and compensates the thermal error of five-axis machine tools. The thermal error modeling technology of five-axis machine tools mainly includes the selection of thermal key points and the establishment of thermal error model.

In this paper, the thermal error modeling of five-axis machine tools is mainly based on the five-point test method to realize the calibration of spindle thermal error, to achieve the prediction of spindle thermal error.

## 2. Thermal Error Model Construction of Spindle on Five-Axis Machine Tools

The overall process of thermal error model construction of spindle on five-axis machine tools is shown in Figure 1. It includes three parts, which are thermal error analysis, thermal characteristics testing, and thermal error modeling. In the thermal error analysis, the machine structure and drive chain are analyzed. It selects reasonable temperature measurement points. The five-point test method is used to measure the thermal error of the machine tool spindle. In the process of thermal characteristics testing, the experimental platform including temperature acquisition and thermal error data acquisition is built. According to the GSK988 data frame format, the intelligent temperature acquisition module transmits the float type temperature data to the host computer through TCP/IP communication protocol. To complete the data acquisition simultaneously, the eddy current data is collected using the NI6356BNC acquisition card. In the process of thermal error modeling, it mainly includes temperature key point selection, temperature and error data synchronization processing, and the use of multiple linear regression models to complete the construction of the thermal error model.

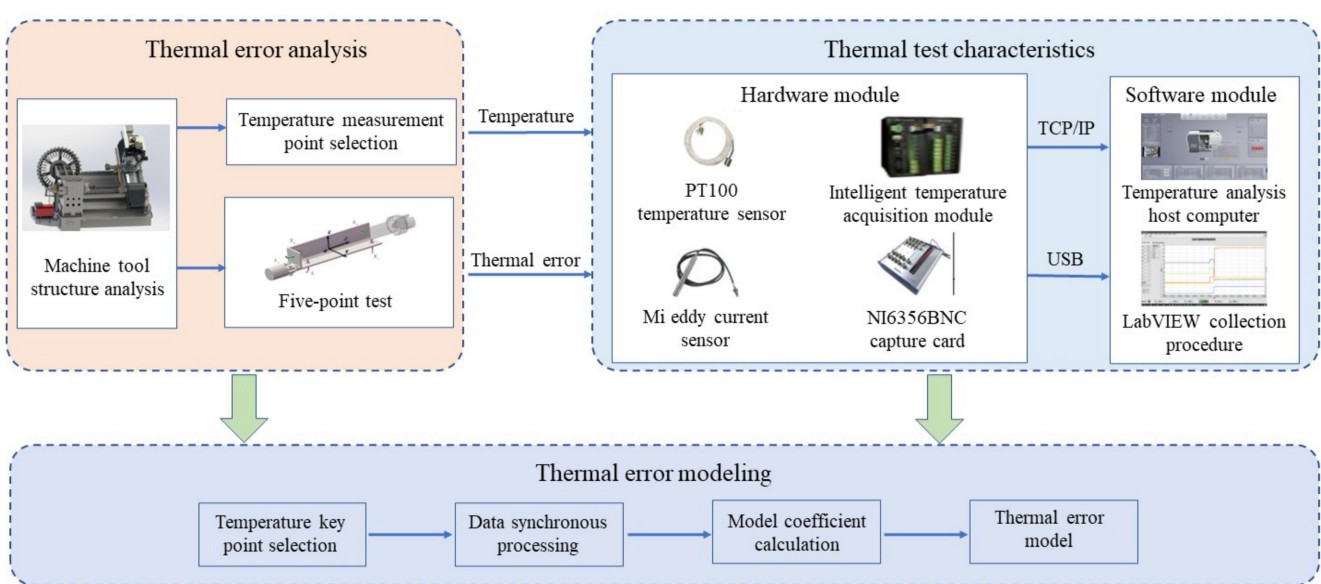

**Figure 1.** The overall flow of the thermal error modeling of the machine tool spindle.

The structure of the milling spindle of the machine tool is shown in Figure 2. The transmission chain of the S-axis (i.e., second spindle or milling spindle) is distinguished by yellow marks. The S-axis motor is fixedly connected with the bevel gear through the motor shaft to realize commutation. The rotating motion is transmitted to the central shaft through the synchronous pulley. The end of the central shaft is fixedly connected with the

bevel gear to realize a 90° commutation. It transfers the rotating motion to the tool handle to realize the milling function. There are two pairs of bevel gears, three rotating central shafts, one synchronous belt pulley, and several bearings on the whole motion chain. The motion chain is relatively complex.

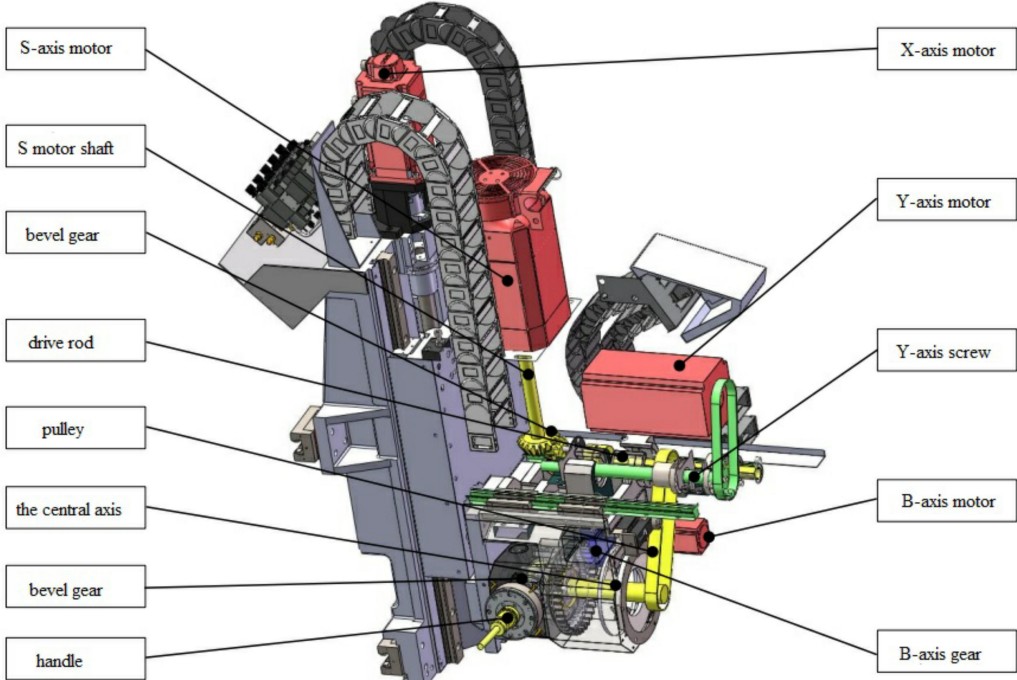

**Figure 2.** Milling spindle drive chain of the machine.

### 3. Data Collection

To accurately measure the overall temperature field of five-axis machine tools, it is first necessary to determine the installation location and number of temperature sensors. In theory, when the temperature field distribution of the spindle system is not clear, it should arrange temperature sensors on the surface. However, the arrangement of a large number of temperature sensors will certainly increase the cost. The subsequent temperature data processing becomes more complex. Too little temperature point arrangement will lose part of the important information. The temperature field cannot truly be reflected. Therefore, it is very important to determine the location and number of temperature measurement points, which will affect the subsequent modelling results. Based on the above analysis of machine tool thermal deformation, the temperature points should be arranged in the thermal displacement information chain components. Combined with actual engineering experience, the arrangement of temperature sensors should meet the following principles.

(1) Heat generated from heat sources is the most important cause of thermal deformation of machine tools, so the location of heat sources should be considered in the temperature point arrangement, such as motors, gears, and bearings.

(2) For the complex location of heat source distribution, it should be as much as possible to arrange the sensor. The use of symmetrical arrangement is to facilitate the subsequent analysis of the temperature gradient. In the machine bed guide, slide, saddle, and column, sensors are symmetrically arranged.

(3) The ambient temperature has a certain influence on the overall expansion of five-axis machine tools. The temperature rise of each temperature point is the basis. After the machine reaches thermal equilibrium, the ambient temperature is a major factor in its influence. Therefore, the ambient temperature measurement points should be arranged. In this experiment, due to the backside of five-axis machine tools for automatic induction switch door, external cold and hot air will also have an impact on

the ambient temperature around five-axis machine tools. Therefore, the switch door should also be arranged with temperature measurement points.

Comprehensive above principles, the initial selection of 19 temperature measurement points as shown in Table 1, of which the ambient temperature uses two temperature measurement points, each temperature sensor installation location as shown in Figure 3.

**Table 1.** Temperature sensor installation location summary table.

| Range | Sensor Installation Location | No. | Range | Sensor Installation Location | No. |
|---|---|---|---|---|---|
| S-axis | S-axis center shaft rear end cover | T1 | X-axis | X-axis saddle lower right | T11 |
| | S-axis cutting fluid integration block | T2 | | S-axis and X-axis motor flange | T12 |
| Bed | Z-axis upper guide left | T3 | Y-axis | Y-axis ball screw support seat | T13 |
| | Z-axis lower guide left | T4 | | Y-axis motor flange | T14 |
| | Z-axis upper guide left front | T5 | | Y-axis column left side | T15 |
| | Z-axis upper guide right front | T6 | | Y-axis column right | T16 |
| | Right end of Z-axis guide | T7 | B-axis | B-axis motor flange | T17 |
| X-axis | X-axis saddle top left | T8 | Environment | Machine top beam | T18 |
| | X-axis saddle lower left | T9 | | Outside of machine near switch door | T19 |
| | X-axis saddle upper right | T10 | | | |

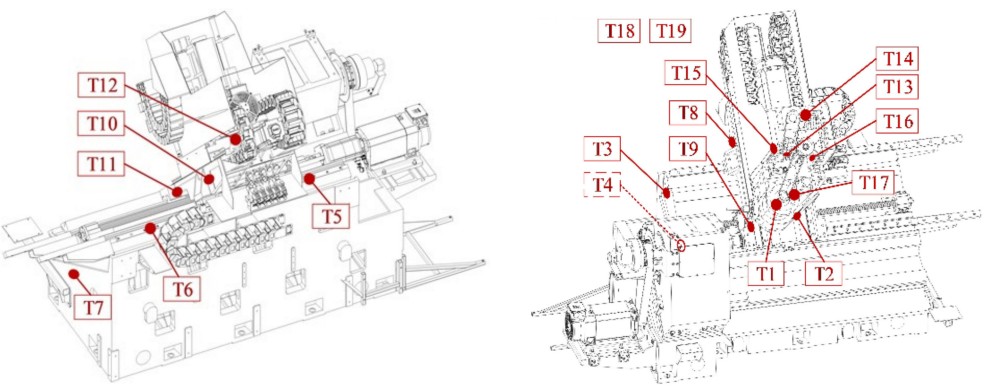

**Figure 3.** Temperature sensor installation location diagram (dashed lines indicate masking).

## 4. Data Analysis

After filtering the collected temperature and eddy current data and aligning the data based on time, the machine temperature data and thermal deformation data at different speeds of 1000 r/min and 2000 r/min were obtained as shown in Figures 4 and 5 below.

From Figures 4 and 5, it can be seen that the temperature change trend of the machine under different speeds is basically the same, the temperature of each measurement point tends to stabilize with time, in which the temperature around the slide plate generally changes drastically compared with other parts of the temperature. When the machine speed is 1000 r/min, the highest temperature of T1 sensor is about 40 °C. When the machine speed is changed to 2000 r/min, the highest temperature of T1 sensor is about 47 °C. The temperature rise of the machine as a whole is greater than the experimental data at low speed due to the increase of heat generation in all parts.

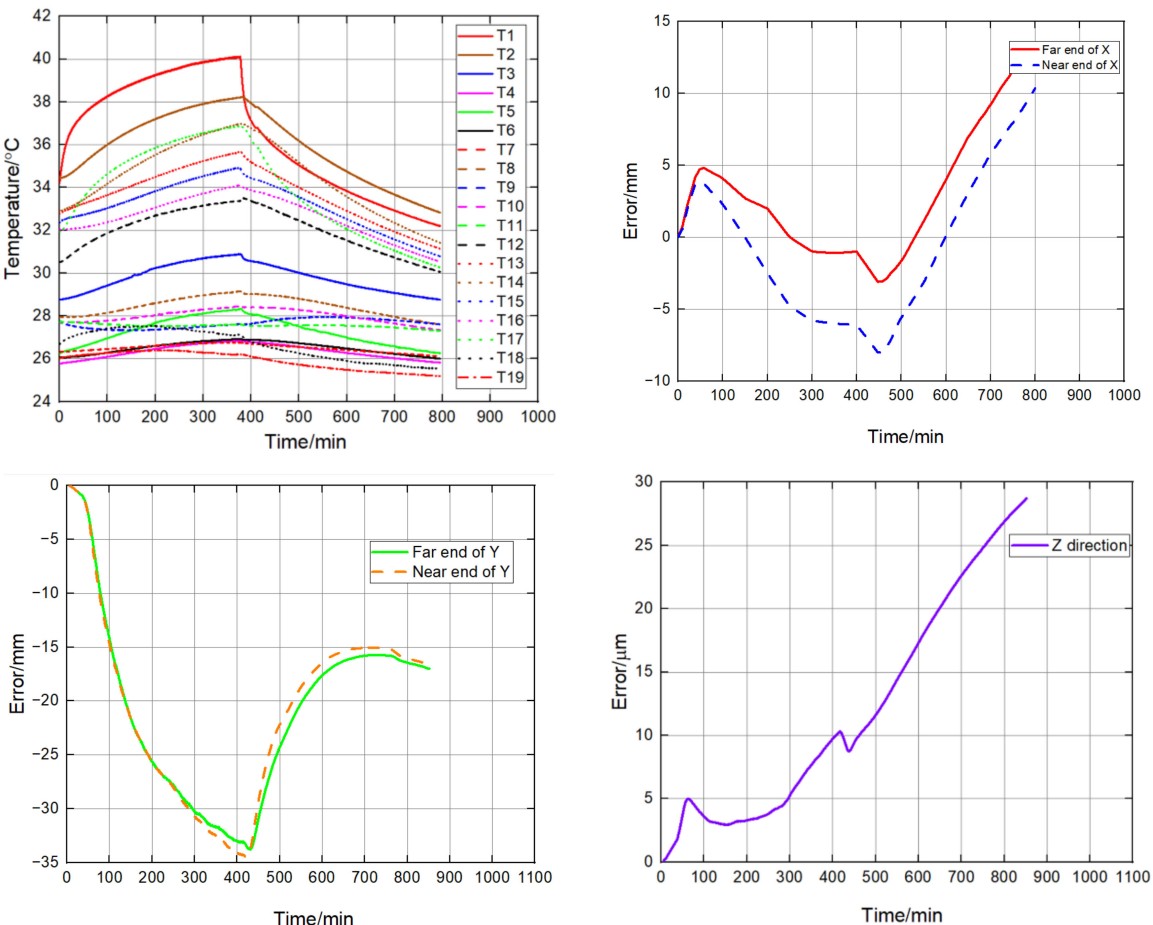

**Figure 4.** 1000 r/min spindle box temperature and heat deformation curve.

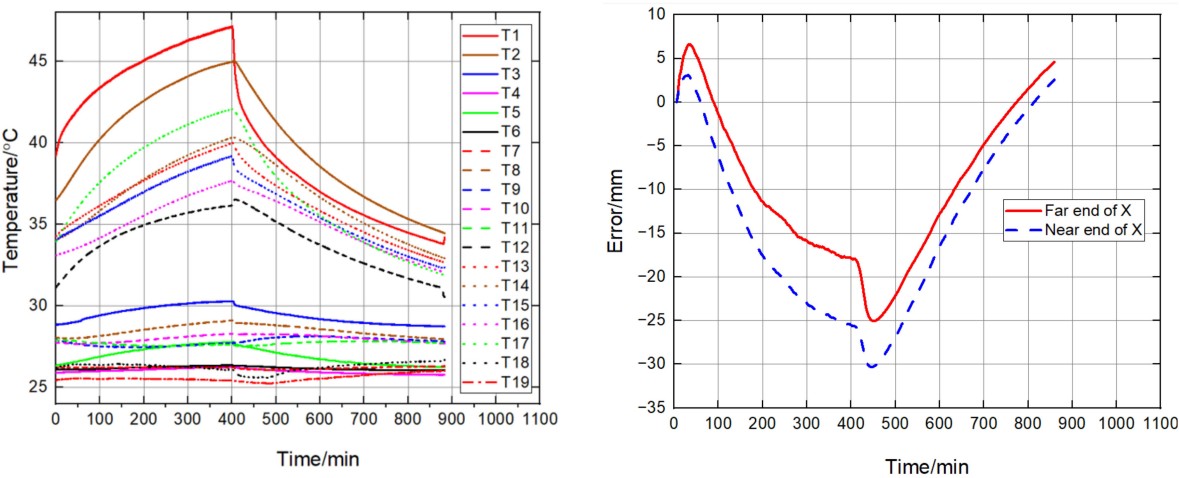

**Figure 5.** 2000 r/min spindle box temperature and heat deformation curve.

As the eddy current sensor measurement range is 0–1 mm, it is usually adjusted to the middle of the measurement range before measurement to facilitate the expansion of the measurement range as much as possible when the direction of the thermal error change is unknown, and its measurement value is the relative displacement from the end face of the eddy current sensor to the position of the tangent plane of the detector bar, and the initial position of each sensor is not consistent, so it is shifted to 0 μm scale for data processing. The data were therefore shifted to the 0 μm scale during data processing. As

can be seen from the figure, the trends of X- and Y-direction thermal errors are basically the same at different rotational speeds, but the difference lies in the inconsistent range of variation. For the X-direction thermal error, as shown in Figure 4, it first increases in the positive direction with the temperature change, and then decreases uniformly in the negative direction after reaching a certain magnitude, until it reaches the thermal equilibrium state with a small change rate. For the thermal error began to have a trend in the positive direction, it is speculated that the X direction is a result of the combined superposition of various components. The thermal deformation obtained from the analysis is in the opposite direction, which is a result of the inconsistent temperature rise rate. It is also possible that in the cold start process, the center shaft, bearings, gears temperature rise faster than the slide components, there is a trend of expansion along the normal direction of the spindle contact surface, in the error measurement diagram shows the trend of increase in the positive direction; as time goes by the heat deformation of the slide in the negative direction gradually increases, showing the overall X-direction thermal error in the negative direction. y-direction thermal error mainly shows an increase in the negative direction until the thermal equilibrium is achieved. For the Z-direction thermal error, the pattern is different between 1000 r/min and 2000 r/min at a cold start, and it is guessed that there is no regularity in the data variation between the central shaft and S-axis due to the bevel gear meshing clearance and other factors.

The final model of thermal error of the X-axis under different working conditions is obtained as follows, which needs to be inverse in the actual compensation process [24].

$$E_x(T) = (44.8120 + 4.8429(T_5) + 0.3919(T_8) - 7.2261(T_{12}) - 0.2493(T_2)/1000(\mu m) \quad (1)$$

The predicted residuals of the model are shown in Figure 6, the maximum value of residuals is about 7.4, which is much smaller than the error range and more uniformly distributed, but the residuals are larger in the process of machine start-up from a cold state, and the model prediction results have phase overrun.

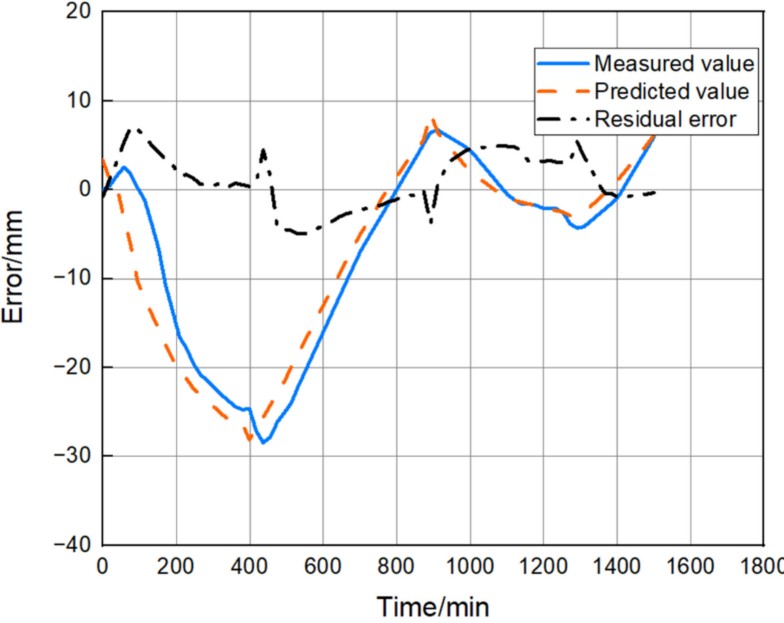

**Figure 6.** X-directional thermal error multiple linear regression model predicted values.

The calculated results of each performance evaluation index are shown in Table 2. The prediction ability of the model is about 77%, and the coefficient of determination is close to 1, indicating that the current model has a certain prediction ability. Compared with LSTM, the prediction accuracy is improved by 5.08%.

**Table 2.** Model fitting performance evaluation indexes.

| Method | $|e_i|_{max}$ Residual Maximum | *MAE* Means Absolute Error | *RMSE* Root Mean Square Error | $R^2$ Coefficient of Determination | $\eta$ Predictive Ability |
|---|---|---|---|---|---|
| LSTM | 7.1256 | 2.0134 | 2.2103 | 0.9013 | 0.7202 |
| This work | 7.4179 | 2.2649 | 2.7914 | 0.9454 | 0.7710 |

## 5. Conclusions

In this paper, the thermal error method of five-axis machine tool is proposed and validated by taking Shenyang machine tool HTM50100 turning and milling machine tool as an example. In the process of thermal error modeling, we select the temperature key point and analyze the collected data. Finally, we evaluate thermal error model. The method is verified by an experiment. The experiment results show that the method is highly accurate, fast, and easy to use.

Meanwhile, by evaluating the method based on multiple linear regression, the predictive ability of the model is about 77%. Compared with LSTM, the prediction accuracy is improved by 5.08%. Therefore, it provides a theoretical basis and practical method for the measurement method of thermal errors of machine tools.

**Author Contributions:** Funding acquisition, Q.L.; Resources, Q.L.; Software, Y.L.; Validation, D.L.; Visualization, D.L.; Writing—original draft, H.T.; Writing—review & editing, Y.L. All authors have read and agreed to the published version of the manuscript.

**Funding:** This research was supported by 2021 Graduate Innovation Fund Project of China University of Geosciences, Beijing: YB2021YC020.

**Institutional Review Board Statement:** Not applicable.

**Informed Consent Statement:** Not applicable.

**Data Availability Statement:** Not applicable.

**Conflicts of Interest:** The authors declare no conflict of interest.

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
