# Peer review of "Thermal Error Analysis of Five-Axis Machine Tools Based on Five-Point Test Method"

_lubricants, doi:10.3390/lubricants10060122_

Round 1
Reviewer 1 Report
Short but interesting paper.
Let´s help to improve it:
- Figure 1 is small and poor.
- Literature: the papers are OK, but you missed several Good leading recent Works: The group by Prof K Weneger was very active as well, see Autonomously triggered model updates for self-learning thermal error compensation. They work for milling centres and turn-milling machine improvement. They even worked on Cutting Fluid Influence
on Thermal Behavior of 5-axis Machine Tools
Thermal models and compensation are two branches of the same topic. You are in the experimental data recording, useful for checking previous models. Gantry machines are well studied, better that turn-mill or mill-turn platforms. - A method for thermal characterization and modelling of large gantry-type machine tools, The International Journal of Advanced Manufacturing Technology 62 (9), 875-886 this is a recent approach and the way to deal with non-linear problems. And the method
was in the top journal in International Journal of Machine Tools and Manufacture 75:100 - 108 . The idea is that a parametric state-space representation was selected as model architecture, providing multiple inputs and outputs capability and a compact
formulation that takes into account previous thermal states of the machine. Inputs for the model are spindle speed and temperatures of main motor gearbox and room air. Outputs are the estimations of the thermal drift of the machine tool centre point along the three axes in different positions within the working volume. - You can do better with the basic state of the art.
- The best part of the work is about results, useful for others: After filtering the collected temperature and eddy current data and aligning the data based on time, the machine temperature data and thermal deformation data at different speeds of 1,000 r/min and 2,000 r/min were obtained.
- I suppose the reason to submit to Lubricants is the temperature values of coolants and lubrication oils, please define it better.
- T1 sensor….it does not depend on speed?
- Conclusions: you need a double space one, give the main points separately. You can write them better.
- The machine brand…ha it any value?
- CNC usually correct thermal deformations, what about this point here.
I think you need to extend the scope of the paper. Results are good, but a deep literature review would help you to define next topics. I see you have Chine and other works at hand, but thermal models are well-known in other parts of the world.

Author Response
Dear Review,
We have finished our revisions according to your advice, thank you very much.

Reviewer 2 Report
The research is not clear enough about the technique used to solve the problem. The paper is short and i think it need to rewrite.
Author Response

(The authors gave the same response as above.)

Reviewer 3 Report
In this paper, authors proposed the accuracy of five-axis machine tools as a key performance indicator and analyzed various 8 error sources of high precision five-axis machine tools, thermal and geometric errors etc. However, this paper is needed to be modified and clarified for several points:
1. The Abstract and Conclusion doesn’t reflect the contents and results of this paper and needed to be revised for dressing the highlights of this paper.
2. The reference didn’t show some other relative papers about the thermal analysis in machine tools. It is suggested to cite some papers in 5 years about the simulation control, compensation or methods relative to the heat caused from the machining or spindle in machine tool as following:
l A Linear Regression Thermal Displacement Lathe Spindle Model
l Adaptive sliding mode control of a high-precision ball-screw-driven stage
l Numerical computation and nonlinear dynamic analysis of ultrasonic cutting system
3. In this paper, authors provide the initial selection of 19 temperature measurement points as shown in Table 1 and the evaluation of the position of the measurement should be discussed and proved in the analysis.
4. This paper applied experiment to show the thermal effect, but didn’t propose any theoretical results to compare the experimental data. If it is possible, please show the comparison with experiment and theoretical results. Nowadays, some commercial software can be available to use for verification of calculation and may authors provide the software simulation results to consider the thermal influence.
5. Did authors consider the couple effect caused by environmental factor like temperature and especially the displacements will be influenced by thermal effect? FEM is a very popular method to deal many kinds of problem in different fields and also can be analyzed for physical coupled phenomena problems.
In summary, the contribution and the results of this manuscript are interesting and it is suggested to revise this paper and please provide some data relative to the experimental analysis proof for further reference.
Author Response

(The authors gave the same response as above.)

Round 2
Reviewer 1 Report
Ok
Reviewer 2 Report
I think the paper is not in the topics of the journal
Reviewer 3 Report
This paper is revised and suggested to accept for further publication!